# Effects of Gap Size on Natural Regeneration in *Picea asperata* Forests of Northern China

**Xin Yang, Jiajing Li, Niqiao Fan, Yiwen Wang and Zhidong Zhang \***

College of Forestry, Hebei Agricultural University, Baoding 071000, China; 17673147206@163.com (X.Y.); 15131780192@163.com (J.L.); fanniqiao@163.com (N.F.); yiwenw123@163.com (Y.W.)
\* Correspondence: zhangzd@hebau.edu.cn

**Abstract:** Our study aimed to assess the impacts of varying forest gap sizes on the density, growth, and spatial patterns of seedlings and saplings in spruce (*Picea asperata*) forests in the Saihanba region, Hebei Province, China. Twenty-four forest gaps were surveyed and categorized into six classes based on the gap size. A one-way ANOVA was used to compare differences in the density, height, and ground diameter of seedlings and saplings among six gap classes. Ripley's *K* function was used to explore the spatial patterns of regeneration establishment in each class. The findings of our study indicated that the forest gap size did not significantly influence the density of seedlings or the ground diameter growth of saplings, whereas it significantly influenced the height growth of saplings. In smaller gaps, natural regeneration occurred primarily in the gap edges. As the gap size increased, the natural generation began to shift from the edge areas to the gap centers. Large forest gaps had the highest percentages of random distribution patterns across all spatial scales. Aggregated distributions were observed at distances less than 1 m in all gap size classes, whereas uniform distributions tended to occur in the small gaps at distances of 2–4 m. Our findings indicated that larger forest gaps, ranging from 60 to 120 m$^2$, were more conducive to spruce regeneration. The results can inform the development of targeted strategies for understory afforestation and the artificial promotion of natural regeneration in spruce forests.

**Keywords:** forest gaps; natural regeneration; spatial distribution patterns; *Picea asperata*; Saihanba

## 1. Introduction

Natural regeneration serves as the foundation for the maintenance the community composition, structural stability, and survival and reproduction of forest populations [1]. In recent years, forest management has evolved from a dominant function to a multifunctional approach. Researchers have increasingly emphasized the role of natural forces in stabilizing forest ecosystems [2,3]. As a result, the focus has shifted towards natural regeneration processes, with particular attention to the role of forest gap disturbances in promoting the regeneration of forests in their natural state [4–6].

British ecologist Watt first introduced the concept of forest gaps in 1947 [7]. A forest gap refers to an area within the forest where the canopy is discontinuous due to tree mortality. Forest gaps can be categorized into two types: extended forest gaps and canopy gaps [8]. Forest gaps, which are characterized as small-scale disturbances occurring within the forest, play a vital role in maintaining forest regeneration and significantly impact plant community diversity [9]. Variations in stand conditions and light availability, as well as biotic and abiotic factors, across different locations within forest gaps contribute to spatiotemporal changes in forest communities. These changes exert substantial impacts on seedling emergence, survival, establishment, and growth [10,11].

Forest gap disturbance has garnered significant research attention in recent years due to its pronounced impact on the natural regeneration pattern. This impact manifests through alterations in the survival and reproduction conditions of various species [12].

Among the influential factors in forest gaps, the gap size and within-gap position are key determinants affecting tree species regeneration [13]. Changes in gap size and within-gap position initially result in shifts in light conditions within the stand, subsequently influencing the spatiotemporal distribution patterns of other microclimatic factors [14]. Research has demonstrated that the forest gap size primarily affects the natural regeneration by influencing photosynthetically active radiation [15]. A previous study revealed a positive correlation between the density of *Pinus thunbergii* seedlings and the gap size [6]. In contrast, as the gap size decreased, the height, ground diameter, crown length, and crown width of *Quercus mongolica* and *Fraxinus mandschurica* saplings were significantly reduced in a natural secondary forest in the Changbai Mountains [16]. However, it is worth noting that there exists a threshold value for the effect of the forest gap size on seedling growth, with seedling regeneration being optimal under medium-sized forest gaps, while excessively large gaps hinder seedling regeneration [17]. Additionally, a study observed that the maximum values of the basal diameter and tree height of red pine (*Pine koraiensis*) and spruce saplings appeared at different within-gap positions as the gap size varied in a spruce–fir mixed stand in the Changbai Mountains [18]. However, the maximum height and basal diameter of *Pinus sylvestris* seedlings were not related to the gap size [19]. Different tree species exhibit varying requirements for gap size. Shade-intolerant or early successional species typically occupy larger gaps [20–22], requiring a minimum gap of 1000 m$^2$ or 400 m$^2$ for regeneration [23,24]. Conversely, shade-tolerant or late successional species regenerate and grow better in smaller or older gaps [25]. The microenvironmental heterogeneity within the forest gap significantly influences the distribution of tree species seedlings and saplings. For example, in a study on *Fagus longipetiolata*, it was observed that photosynthetically active radiation displayed greater variability in large forest gaps than in small gaps, leading to heterogeneity in seedling distribution in large gaps [26]. Similarly, the maximum densities and heights of *Pinus tabuliformis* seedlings and saplings were found in the northeastern part of the gap [27]. Despite the crucial role of forest gap disturbance in promoting forest regeneration and increasing species diversity, its effects on the regeneration process are multifaceted and influenced by factors such as the species' biological characteristics, regeneration modes, resource availability, and species' relative positions within the gaps [28,29].

The spruce (*Picea asperata*) forest is one of the major forest types in Northern China and plays an important role in soil and water conservation. Natural regeneration is the main means of spruce population propagation, with forest gaps serving as the main sites for this regeneration. To gain deeper insights into the effects of forest gap characteristics on the density, growth, and spatial patterns of spruce regeneration, this study conducted an investigation encompassing 24 forest gaps of varying sizes. We employed various analytical methods, including analysis of variance (ANOVA), kernel density estimation, and point pattern analysis. Specifically, the study mainly addressed the following three questions: (1) how does the density and growth of spruce regeneration individuals vary with the forest gap size? (2) is there a significant variation in the spatial distribution of spruce regeneration among different-sized forest gaps? and (3) is the spatial pattern of spruce regeneration spatially scale-dependent within the forest gaps? The results of the study are expected to provide a scientific basis for an enhanced understanding of the natural regeneration associated with spruce forest gaps.

## 2. Materials and Methods

### 2.1. Site Description

The study was conducted in the Saihanba Mechanical Forest Farm, located in Chengde City, Hebei Province, China (42°22′42.31″ N, 116°53′117.31″ E). It covers an area of 933.3 km$^2$ with elevations ranging from 1500 to 1939.6 m. It has a cold temperate continental monsoon climate. The average annual temperature is −1.40 °C and the extreme maximum and minimum temperatures are 30.9 °C and −42.8 °C, respectively. The average annual precipitation is 438 mm, contrasted with evaporation of 1230 mm. Furthermore,

the average frost-free period lasts around 60 days annually [30]. Spruce forest is one of the major forest types in the study area. The dominant shrub species in spruce forest include *Padus racemosa*, *Ribe nigurm*, *Lonicera chrysantha*, *Malus baccata*, *Catalpa ovata*, and *Rosa davurica*. The herbaceous species in spruce forest mainly include *Carex ussuriensis*, *Potentilla flagellaris*, *Polygonatum humile*, *Iris ventricosa*, *Vicia cracca*, *Viola prionantha*, and *Pyrola calliantha*.

### 2.2. Data Collection

A total of 24 forest gaps were enclosed by outlining the canopy projections of the border trees (Figure 1). For each gap, we recorded the species name, diameter at breast height (DBH), height, and location coordinates (X, Y) of the border trees. The center of gravity of the connecting line between the border trees was used as the coordinate origin (0, 0) for each gap. The forest gaps were divided into six classes based on the gap area: Class I (4–8 m$^2$), Class II (8–12 m$^2$), Class III (12–16 m$^2$), Class IV (16–20 m$^2$), Class V (20–60 m$^2$), and Class VI (60–120 m$^2$).

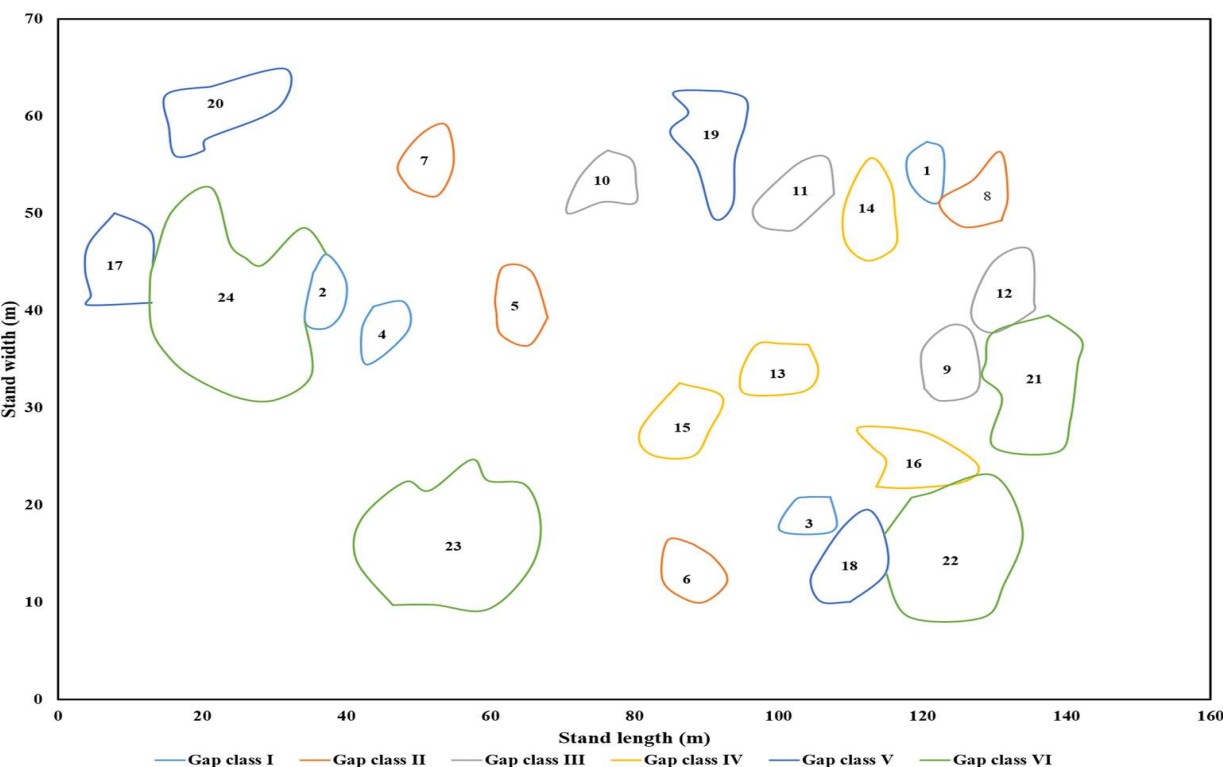

**Figure 1.** The locations of gaps in spruce forest.

All forest regeneration individuals within the forest gaps were surveyed. The survey indicators included the relative coordinates, height, and ground diameter for each individual. Regeneration individuals were categorized based on the ground diameter (GD) into seedlings (GD < 20 mm) and saplings (20 mm < GD < 50 mm) (Table 1).

**Table 1.** The basic characteristics of forest gaps.

| Gap Classes | ID | Area/m$^2$ | Length/m | Mean Height of Border Tree/m | Mean DBH of Border Tree/cm | Border Tree Species |
|---|---|---|---|---|---|---|
| I | 1 | 6.55 | 10.25 | 17.30 | 24.10 | YS, BH |
|  | 2 | 7.23 | 10.78 | 17.40 | 22.20 | YS |
|  | 3 | 5.94 | 10.06 | 18.65 | 28.82 | YS |
|  | 4 | 4.97 | 9.64 | 15.53 | 21.39 | YS |
| II | 5 | 11.28 | 16.75 | 17.30 | 23.40 | YS |
|  | 6 | 10.06 | 12.68 | 18.20 | 21.70 | YS |
|  | 7 | 8.17 | 11.74 | 11.51 | 13.67 | YS |
|  | 8 | 9.47 | 14.12 | 15.22 | 20.58 | YS, BH |
| III | 9 | 14.70 | 15.99 | 16.70 | 20.70 | YS |
|  | 10 | 15.86 | 16.73 | 18.00 | 27.72 | YS |
|  | 11 | 13.71 | 14.45 | 13.23 | 20.31 | YS |
|  | 12 | 12.04 | 22.96 | 19.30 | 25.90 | YS, LYS |
| IV | 13 | 16.85 | 17.22 | 16.70 | 20.40 | YS |
|  | 14 | 16.40 | 15.38 | 12.50 | 17.30 | YS |
|  | 15 | 17.72 | 22.06 | 16.30 | 22.52 | YS, SY |
|  | 16 | 17.41 | 16.02 | 16.82 | 25.03 | YS |
| V | 17 | 28.63 | 22.17 | 17.57 | 26.00 | YS |
|  | 18 | 30.16 | 25.00 | 16.63 | 27.54 | YS |
|  | 19 | 37.18 | 25.36 | 20.30 | 30.30 | YS, BH |
|  | 20 | 22.72 | 19.71 | 19.10 | 26.93 | YS |
| VI | 21 | 68.43 | 35.39 | 16.30 | 27.30 | YS |
|  | 22 | 82.78 | 34.24 | 19.26 | 28.83 | YS |
|  | 23 | 103.21 | 41.39 | 18.57 | 28.00 | YS |
|  | 24 | 114.16 | 47.97 | 17.60 | 26.00 | YS, BH |

YS: *Picea asperata*; BH: *Betula platphylla*; LYS: *Larix principis-rupprechtii*; SY: *Populus davidiana*. I: forest gap class I (4–8 m$^2$), II: forest gap class II (8–12 m$^2$), III: forest gap class III (12–16 m$^2$), IV: forest gap class IV (16–20 m$^2$), V: forest gap class V (20–60 m$^2$), VI: forest gap class VI (60–120 m$^2$). The same below.

*2.3. Statistical Analysis*

2.3.1. Differences in Density, Height, and Ground Diameter

One-way ANOVA with the least significant difference (LSD) test and multiple comparisons were performed to quantitatively assess the responses of the spruce regeneration density, height, and ground diameter to various gap size classes. Prior to statistical analysis, all data were standardized to meet normality requirements. Statistical significance was tested at $p < 0.05$. The analysis was conducted in the SPSS 27.0 software.

2.3.2. Kernel Density Estimation

Kernel density estimation (KDE) is a widely used non-parametric statistical method. It utilizes a quadratic kernel function to model a smooth surface at each point to calculate the size of the unit area [31]. The formula for kernel density estimation is as follows:

$$\hat{f}_h(x) = \frac{1}{nh} \sum_{i=1}^{n} k\left(\frac{x - x_i}{h}\right) \tag{1}$$

where $k\,(.)$ is a non-negative function characterized by an integral equal to 1 and a mean of 0; $n$ is the number of data points ($x_i$); $h$ is the search radius.

2.3.3. Spatial Pattern Analysis

The pair correlation function $g(r)$ was used to analyze the spatial patterns of regeneration individuals within spruce forest gaps. *Ripley's K* function is a common function for the analysis of point patterns, but the $g(r)$ function can effectively avoid potential errors caused

by the "virtual aggregation" of the $K(r)$ function [32,33]. The relationship between the $K(r)$ function and the $g(r)$ function can be expressed by the following formula:

$$g(r) = \frac{dK(r)}{2\pi r d(r)} \tag{2}$$

The scale $r$ is used as the horizontal coordinate and the upper and lower envelopes represent the vertical coordinates in the analysis. If the actual value of the $g(r)$ function falls within the range defined by the upper and lower envelopes, it indicates a random distribution of regeneration individuals. If the actual value of the $g(r)$ function is above the upper envelope, it suggests an aggregated distribution, while a value below the lower envelope indicates a uniform distribution.

ArcMap 10.7 was used to compute the kernel density of regeneration individuals in each forest gap, and then to generate the density distribution map for spruce regeneration individuals. The spatial pattern of spruce regeneration individuals was analyzed using the ecological software Programita 2018.

## 3. Results

### 3.1. Differences in Density, Height, and Ground Diameter

There were no significant differences in spruce seedlings, saplings, and total regeneration density among different gap size classes, with variations ranging from 0.32 to 11.39 stems/m$^2$, 0 to 0.82 stems/m$^2$, and 0.38 to 11.65 stems/m$^2$, respectively ($p = 0.63$). Seedlings and overall regeneration densities were highest in class VI (60–120 m$^2$) and lowest in class III (12–16 m$^2$), while the sapling density peaked in class III and was lowest in class II (8–12 m$^2$). It is noteworthy that the seedling density was significantly higher than the sapling density in all gap size classes ($p = 0.0002$).

The heights of spruce seedlings and saplings exhibited an increasing trend with increasing gap sizes, reaching their highest heights in gap class VI (60–120 m$^2$). No significant impact on the ground diameter growth of spruce regeneration individuals was observed across different gap size classes ($p = 0.18$). However, there was a significant increasing trend in ground diameter with increasing gap sizes ($p = 0.04$). Seedlings had the highest average ground diameter in class III (12–16 m$^2$), while saplings achieved their maximum in class VI (60–120 m$^2$) (Figure 2).

### 3.2. Spatial Distribution of Regeneration Individuals

The distribution of regeneration individuals exhibited distinct patterns in various forest gap size classes. In class I and II forest gaps, the distribution was notably scattered. In contrast, in class III, IV, and V forest gaps, the distribution of regeneration individuals began to show a shift from the edge to the center of the forest gaps. The regeneration individuals in class VI forest gaps showed an aggregated distribution. The maximum density of regeneration individuals was found near the center of small forest gaps in classes I, II, and III. However, in medium and large forest gaps (classes IV, V, and VI), the maximum density of regeneration individuals exhibited a broader distribution, spanning from the forest gap edges to the central region (Figure 3).

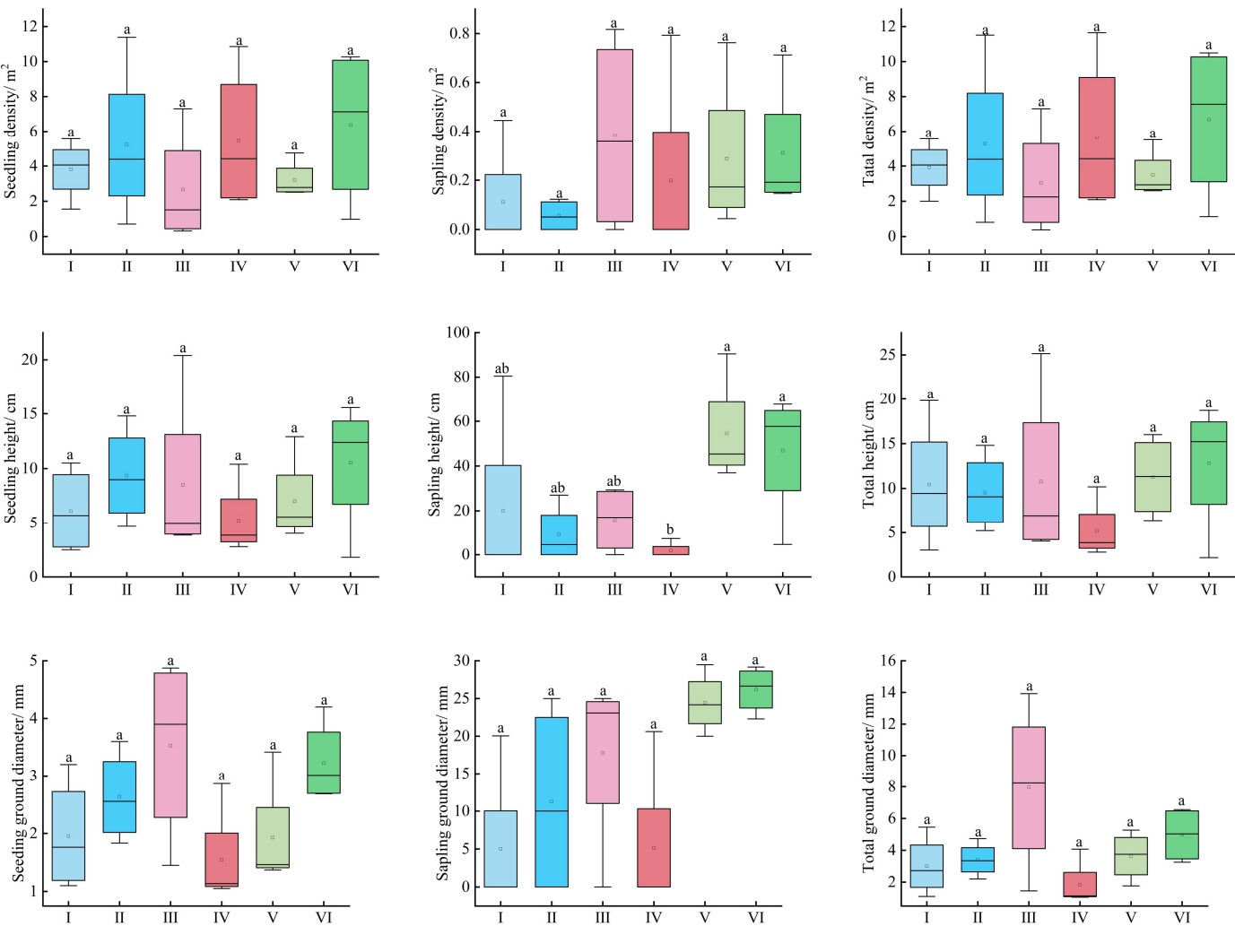

**Figure 2.** Differences in density, height, and ground diameter of spruce regeneration individuals (seedlings, saplings, and total) among different forest gap sizes. Lowercase letters above the standard error bars indicate significant differences (LSD test) among different gap sizes ($p < 0.05$). I: forest gap class I (4–8 m$^2$), II: forest gap class II (8–12 m$^2$), III: forest gap class III (12–16 m$^2$), IV: forest gap class IV (16–20 m$^2$), V: forest gap class V (20–60 m$^2$), VI: forest gap class VI (60–120 m$^2$). The same below.

### 3.3. Point Pattern Analysis of Regeneration Individuals

In all gap size classes, except class II forest gaps, spruce regeneration mainly exhibited a random distribution pattern, which occurred at an average percentage of 57.02% across the examined distances (Figure 4). Large forest gaps had the highest percentages of random distribution patterns across all scales, 73.77% for class V and 60.53% for class VI forest gaps, respectively. Aggregated distributions were observed at distances less than 1 m in all gap size classes, whereas uniform distributions tended to occur in the small gaps (from class I to class IV) at distances of 2–4 m.

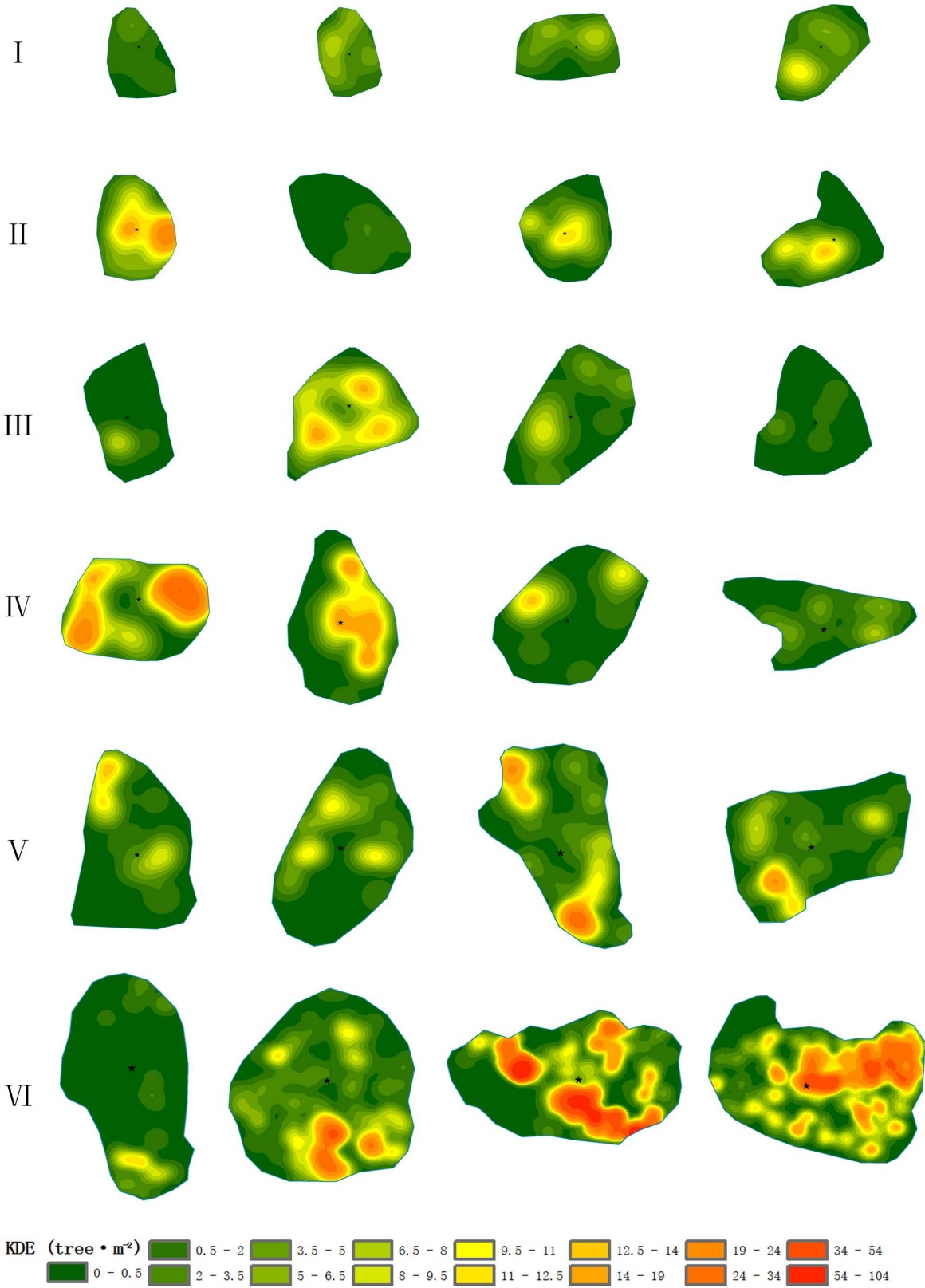

**Figure 3.** Estimation of mean kernel density of spruce regeneration individuals in different gap size classes; ★: The gap center.

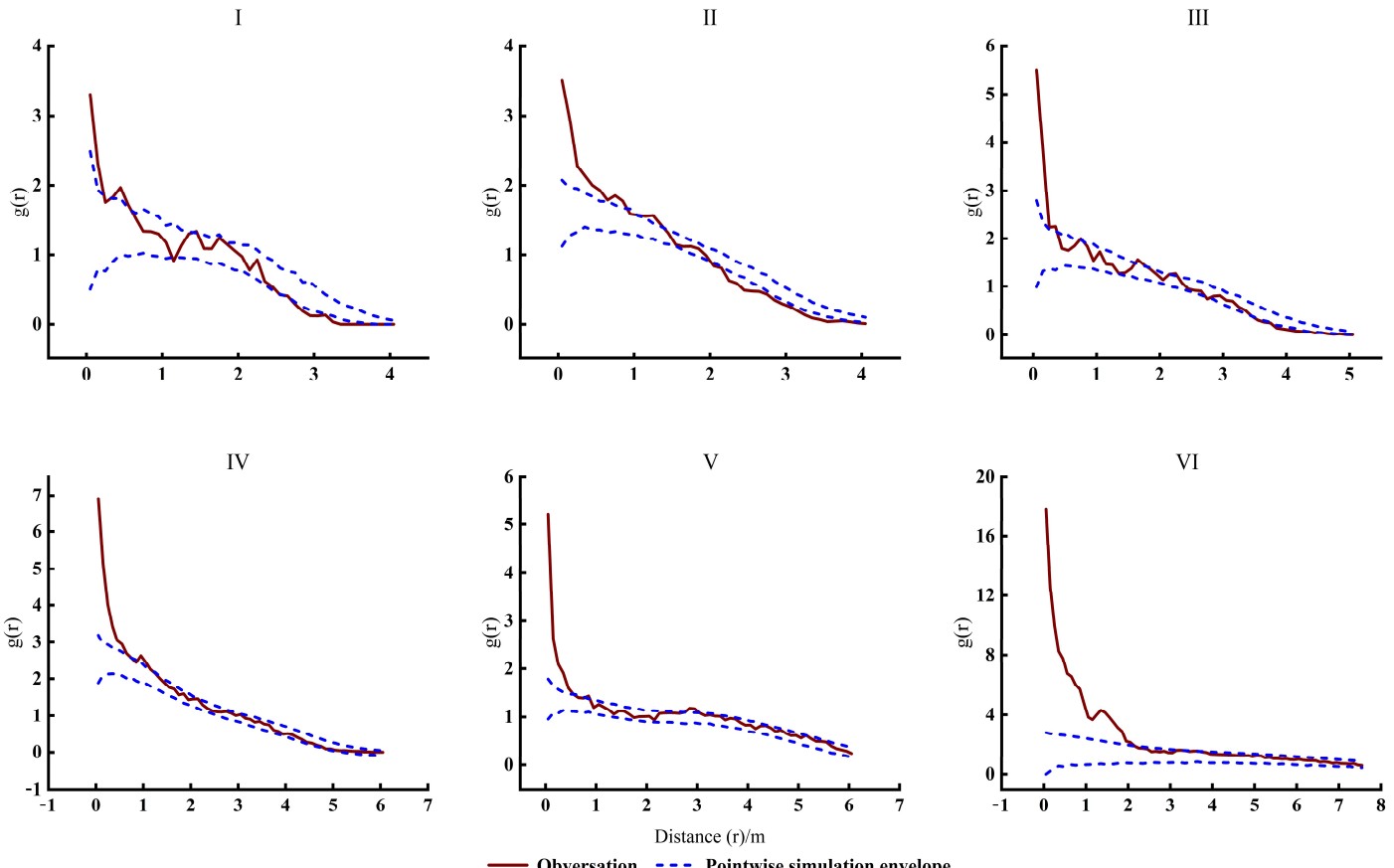

**Figure 4.** Analysis of point patterns for spruce regeneration individuals.

## 4. Discussion

### 4.1. Variations in Density, Height, and Ground Diameter of Regeneration among Gap Size Classes

The gap size did not significantly influence the overall density, height, and ground diameter of spruce regeneration individuals (Figure 2). The densities of spruce regeneration ranged from 3.0 to 7.4 trees/m$^2$ across the six size classes of forest gaps (Figure 2). Interestingly, there was an observable trend of increasing regeneration density with the increase in gap size. The maximum densities of seedlings were observed in class VI gaps, while the maximum density of saplings was observed in class III gaps (Figure 2). These variations in maximum densities across different gap size classes may be attributed to responses to environmental heterogeneity in the different-sized gaps [26]. Similar studies have shown that larch seedlings had maximum densities in forest gaps ranging from 100 to 150 m$^2$ [34], and the sapling densities were highest in medium forest gaps (60~120 m$^2$) [35]. Regarding the growth of regeneration individuals, the gap size had a significant impact only on the height growth of spruce saplings. The sapling height in large forest gaps (classes V and VI) was slightly reduced compared to that in medium-sized gaps (e.g., classes III and IV). The lowest height of saplings was found in forest gap size class IV (Figure 2). This result may be due to the fact that, in gap class IV (16–20 m$^2$), where herbs were more abundant and the herb height was higher than that of saplings, saplings were at a competitive disadvantage. In contrast, gap class VI (60–120 m$^2$), representing a suitable gap microenvironment for regeneration, yielded the highest total density of regeneration, thus conferring a competitive advantage over herbs and resulting in the highest sapling height. Generally, the creation of gaps promotes regeneration growth. However, if the gap diameter exceeds the height of the border trees, it hinders regeneration individuals' growth. This is because such larger gaps contribute to increased ground vegetation cover and microenvironmental changes within the gap, which ultimately inhibit the growth of regeneration individuals during this

phase [26,36–38]. Spruce, characterized by its small seeds, primarily deposits seeds on the forest floor litter. Herbaceous cover was found to influence the survival and establishment of regeneration individuals [39]. Excessive herbaceous cover has adverse effects on germination, establishment, and the spatial distribution of regeneration individuals [40–42]. Spruce is a shade-tolerant tree species that requires increased light during the growth phase of saplings. However, growth slows down when the light intensity exceeds the suitable threshold for spruce saplings [43]. Consequently, the growth of spruce saplings benefits from a larger spatial ecological niche, and, as their competitive ability increases, the impact of shrub and grass vegetation gradually diminishes. This environment is more favorable for the growth of saplings in larger forest gaps. The study also revealed a decrease in the survival rate of spruce regeneration seedlings with increasing age. This suggests that spruce seedlings require proper rooting in the soil for sustained growth. In our study stand, the litter was thicker, leading to a higher mortality rate for spruce seedlings. A higher mortality rate often results from embryonic roots failing to reach the soil and access sufficient nutrients. Measures such as forest fire control and soil turnover can effectively enhance the survival rate of spruce regeneration individuals. Moreover, seed production and quality are guarantees of successful regeneration [44,45]. The seed number and size will influence spruce colonization and seedling success. In our forthcoming study, we will delve deeper into the impact of forest gaps on spruce regeneration, particularly examining how it relates to both seed size and number.

### 4.2. Spatial Distribution of Regeneration Individuals within Forest Gaps

The kernel density map revealed that spruce regeneration was primarily distributed at the edges of forest gaps (Figure 3). As the gap area increased, the regeneration individuals gradually shifted from the edges to the center of the gaps (Figure 3). The shaded areas created by the border trees at gap edges not only offer the spruce regeneration protection but also provide favorable soil conditions and light intensity, thereby promoting seed germination, establishment, and growth. The distribution of regeneration individuals varies across different-sized forest gaps due to differences in the specific microenvironment of different within-gap positions [15]. In forest gaps, the competition for light is more intense. Larger forest gaps exhibit higher light intensity compared to smaller gaps. Moreover, an increasing gap area results in microenvironmental gradient changes from the gap edges to the center [46]. These variations in microenvironment contribute to differences in the distribution of regeneration individuals within the gaps. The heterogeneity of the resource environment within forest gaps is influenced by several factors, including the number and height of border trees and the crown inclination of these trees. Ultimately, the regeneration individuals grow in locations where the light source is suitable within the forest gaps [35]. The results of the study are in consistent with the previous findings of Vilhar et al. [26]. They concluded that the center area of the forest gap and the sunlit edge are important areas for the survival and growth of regeneration individuals. However, canopy growth over time will reduce the availability of light at the gap edges [47], thus hindering natural regeneration. Therefore, while specific locations within gaps might be conducive to the early survival of seedlings, they may not support seedling growth throughout their lifespan [48]. The location within a forest gap holds significant importance in understanding gap dynamics. It directly affects the amount of solar radiation reaching the forest floor, resulting in changes in the microclimate and soil nutrient availability within the forest gaps. These factors, in turn, play a crucial role in natural regeneration [49]. Future studies on spruce regeneration seedlings should primarily focus on examining the relationships between regeneration and important factors, including the light distribution within forest gaps, soil moisture, and temperature, as well as litter depth, etc.

### 4.3. Analysis of the Point Pattern of Regeneration

The spatial distribution pattern of the regeneration individuals mainly showed a random distribution in most forest gaps, becoming more apparent as the forest gap increased

(Figure 4). This pattern was mainly closely related to the degree of heterogeneity in resource distribution [27]. Therefore, we speculate that the random distribution in different-sized gaps was mainly attributed to the lower variability in microenvironmental heterogeneity in spruce forest gaps. Furthermore, the aggregation of spruce regeneration mainly occurred in the scale range of <1 m (Figure 4), which is similar to the findings of Yan et al. [50]. The reason for this phenomenon may be that the clustered regeneration seedlings in the forest gap can shelter each other from external environmental damage and improve their survival chances [51]. The spatial pattern variability of regeneration across different scales is influenced by several factors, such as intraspecific or interspecific competition and abiotic factors like topography and soil conditions [52,53]. Moreover, the distribution pattern of regeneration individuals is strongly influenced by seed dispersal [32], as seed dispersal limitations (seed size, seed predation, and the direction of prevailing winds) lead to heterogeneity in seed germination and seedling growth [54]. Under these heterogeneous conditions, regeneration individuals differentiate, leading to natural thinning, reduced competition, and increased resilience [55]. In this study, we found that the spatial pattern of spruce seedlings was highly scale-dependent. In particular, regeneration individuals at a distance scale of 2–4 m in small forest gaps increased the competition for resources, especially light, water, and nutrients, resulting in repulsion among each other, which in turn led to regular patterns. The varied spatial patterns of regenerating spruce individuals at different spatial scales in different-sized forest gaps are indicative of the interaction among micro-site heterogeneity, seed dispersal limitation, and intra- and interspecific competition. The mechanism behind the spatial pattern formation in spruce regeneration requires further research.

## 5. Conclusions

In this study, we conducted an analysis to investigate the impact of the forest gap size on the density, growth, and spatial distribution patterns of spruce regeneration. The results revealed significant effects of the forest gap size on the growth of spruce saplings. Specifically, we observed a shifting pattern in the distribution of spruce regeneration from the forest gap's edge towards its center as the size of the forest gap increased. This observation suggests that seedlings exhibit a preference for growth at the forest gap's edge, shaded by the forest canopy. Conversely, spruce saplings tended to thrive in the central region of the forest gap, where the light intensity was greater. Spruce regeneration exhibited a stronger random distribution pattern in larger forest gaps compared to smaller gaps. Notably, larger forest gaps, ranging from 60 to 120 $m^2$, were more conducive to spruce regeneration. In the context of sustainable spruce forest management, it is imperative to assess the long-term effects of varying gap sizes, particularly large gaps, on natural regeneration. Forest managers need to monitor the density and growth of regeneration individuals in small and medium-sized gaps in spruce forests for a long time. When deemed necessary, they should also implement manual management strategies such as thinning, pruning, and the clearing of forest ground cover and litter. These measures are crucial in establishing and preserving a favorable microenvironment conducive to spruce regeneration. A comprehensive understanding of the effects of gap size on the density, growth, and spatial patterns of spruce regeneration is essential for forest managers. Such insights can inform the development of targeted strategies for understory afforestation and the artificial promotion of natural regeneration in spruce forests.

**Author Contributions:** Conceptualization, X.Y. and Z.Z.; methodology, software, visualization, X.Y. and J.L.; investigation, data curation, Z.Z. and N.F.; writing—original draft preparation, X.Y. and Y.W.; writing—review and editing, Z.Z. All authors have read and agreed to the published version of the manuscript.

**Funding:** This research was funded by the Hebei Province Forest and Grass Science and Technology Demonstration Project, grant number TG [2022]018, and the Hebei Province Key R & D Program of China, grant number 22326803D.

**Data Availability Statement:** The data presented in this study are available on request from the corresponding author.

**Acknowledgments:** The authors thank to everyone who helped with the field survey and the anonymous reviewers for their valuable comments.

**Conflicts of Interest:** The authors declare no conflict of interest.

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
