# Peer review of "Effects of Gap Size on Natural Regeneration in Picea asperata Forests of Northern China"

_forests, doi:10.3390/f14102102_

Round 1
Reviewer 1 Report
The interesting study supported by detailed multi-size analyses of regeneration density, distribution and biometric features. Most of differences showed in figures are not significant statistically, so drawing final conclusions is difficult. However, general trends similar to others, known from comparable experiments. Between results I found one, very interesting, concerning regeneration height, especially saplings (Fig. 1, middle part). Why the lowest height (statistically significant!) was found in a case of gap size IV? It needs some comment. Maybe (suggestion for the future works) the clearer results can be achieved by minor number of gap size classes?
Some other fine errors or unclear phrases highlighted in the text.
Reviewer 2 Report
This study aims to understand the effects of forest gap on seedlings and saplings in Picea asperata in Hubei province, China. They found that forest gap size does not significantly influence the density and ground diameter of saplings, but influences the height growth of saplings. They also found larger gaps were more suitable for spruce regeneration. Findings from this study can provide insights into forest management. I think this paper is well written and can be published in Forests after addressing the following comments.
Introduction: Line 27-32: I like the way the authors are highlighting the importance of natural regeneration. I think the author also need to mention about seed production as an additional measurement in forest regeneration. There is a recent paper on Nature Communications https://www.nature.com/articles/s41467-022-30037-9 that quantifies seed production across a variety of forest trees. Seed production serves as the foundation for regeneration. Even though this paper does not use data for seeds, it is still important to mention this paper and maybe talk about it a bit in the discussion.
Line 54: maybe also mention the study regions of those papers.
Line 60: same as above, the study regions can be included.
Methods: table 1, I am wondering if it is possible to include a map showing the gap locations, and labeling the tree IDs. The figure can help visualize the study design, just my thought.
Line 125-126: it is important to include the exact p-value.
Line 136: citation on the Ripley’s K function seems to be missing.
Results: Line 153: is regeneration density using Ripley’s K function?
Line 155: it will help the reader if the author can indicate the gap size for each class VI/III. For example, class VI (larger gaps), and class III (smaller gaps).
Fig 1: it could be helpful to include the exact p-value (instead of using p < 0.05).
Fig 2: I think it would be helpful to indicate the gap locations in this figure. Just my thoughts.
Discussion: L196: which figure is these results referring to?
L212: This paper https://doi.org/10.3389/fevo.2021.719141 and this paper: https://doi.org/10.1890/06-1097 also support the statement that the seedling stages seem to be very sensitive to microenvironmental changes north American tree species.
L230: I really like the kernel density map part of this paper. It is very well done.
Conclusion: L291-L292: can be revised to Spruce regeneration exhibits a higher random distribution pattern in larger forest gaps compared to smaller gaps.
Reviewer 3 Report
The article is written carefully, knowing that several work has been published in different journal on the effects of gap size on natural regeneration with some other forest species. Please suggest strategies in the recommendation of the research study to restore the afforestation
Round 2
Reviewer 2 Report
Thanks for addressing my comments. I think this manuscript is ready for publication.